# Compensatory Neuroprotective Response of Thioredoxin Reductase against Oxidative-Nitrosative Stress Induced by Experimental Autoimmune Encephalomyelitis in Rats: Modulation by Theta Burst Stimulation

**DOI:** 10.3390/molecules25173922

**Published:** 2020-08-27

**Authors:** Ivana Stevanovic, Milica Ninkovic, Bojana Mancic, Marija Milivojevic, Ivana Stojanovic, Tihomir Ilic, Maja Vujovic, Mirjana Djukic

**Affiliations:** 1Institute of Medical Research, Military Medical Academy, 11000 Belgrade, Serbia; ivana.stevanovic@vma.mod.gov.rs (I.S.); milica.ninkovic@vma.mod.gov.rs (M.N.); marija.p.milivojevic@gmail.com (M.M.); 2Medical Faculty of the Military Health Department, Ministry of Defense, 11000 Belgrade, Serbia; malicevic.bojana@gmail.com (B.M.); tihomir.ilic@vma.mod.gov.rs (T.I.); 3Institute for Biochemistry, Faculty of Medicine, University of Nis, 18000 Nis, Serbia; stojanovicivana38@gmail.com; 4Institute for Toxicology, Faculty of Medicine, University of Nis, 18000 Nis, Serbia; majavujovic1@gmail.com; 5Department for Toxicology, Faculty of Pharmacy, University of Belgrade, 11221 Belgrade, Serbia

**Keywords:** thioredoxin reductase, oxidative stress, nitrosative stress, theta burst stimulation, experimental autoimmune encephalomyelitis, rats

## Abstract

Cortical theta burst stimulation (TBS) structured as intermittent (iTBS) and continuous (cTBS) could prevent the progression of the experimental autoimmune encephalomyelitis (EAE). The interplay of brain antioxidant defense systems against free radicals (FRs) overproduction induced by EAE, as well as during iTBS or cTBS, have not been entirely investigated. This study aimed to examine whether oxidative-nitrogen stress (ONS) is one of the underlying pathophysiological mechanisms of EAE, which may be changed in terms of health improvement by iTBS or cTBS. Dark Agouti strain female rats were tested for the effects of EAE and TBS. The rats were randomly divided into the control group, rats specifically immunized for EAE and nonspecifically immuno-stimulated with Complete Freund’s adjuvant. TBS or sham TBS was applied to EAE rats from 14th–24th post-immunization day. Superoxide dismutase activity, levels of superoxide anion (O_2_^•–^), lipid peroxidation, glutathione (GSH), nicotinamide adenine dinucleotide phosphate (NADPH), and thioredoxin reductase (TrxR) activity were analyzed in rat spinal cords homogenates. The severity of EAE clinical coincided with the climax of ONS. The most critical result refers to TrxR, which immensely responded against the applied stressors of the central nervous system (CNS), including immunization and TBS. We found that the compensatory neuroprotective role of TrxR upregulation is a positive feedback mechanism that reduces the harmfulness of ONS. iTBS and cTBS both modulate the biochemical environment against ONS at a distance from the area of stimulation, alleviating symptoms of EAE. The results of our study increase the understanding of FRs’ interplay and the role of Trx/TrxR in ONS-associated neuroinflammatory diseases, such as EAE. Also, our results might help the development of new ideas for designing more effective medical treatment, combining neuropsychological with noninvasive neurostimulation–neuromodulation techniques to patients living with MS.

## 1. Introduction

Experimental autoimmune encephalomyelitis (EAE) is an animal model of relapsing-remitting multiple sclerosis (MS). Repeated neuroinflammatory attacks induce demyelination, neuronal damage/loss, reactive gliosis, the formation of sclerotic plaques, and aggravate remyelination. Those processes are oxidative-nitrosative stress (ONS) associated [1].

The mitochondrial respiratory chain is a primary source of reactive oxygen species (ROS) in physiological conditions [2]. The overproduction of ROS and reactive nitrogen species (RNS) mainly arise from activated glial and T cells and overwhelms the antioxidant capacity in EAE neuroinflammation. Enzymatically catalyzed redox reactions deplete the donors of reducing equivalents, such as nicotinamide adenine dinucleotide phosphate (NAD(P)H) and glutathione (GSH). The central antioxidant enzymes that catalyze ROS sequestration are superoxide dismutase (SOD) and catalase. At the same time, glutathione reductase (GR), glutathione peroxidase, and glutathione-S-transferase are crucial enzymes of the glutathione cycle. Thioredoxin (Trx) and thioredoxin reductase (TrxR) make up the thioredoxin system [3]. The engagement and role of the Trx/TrxR system have not been clarified entirely [4,5]. It is known that Trx as an endogenous antioxidant, reduces disulfide forms of proteins. Flavo- and selenoprotein TrxR regenerates the oxidized form of Trx by utilizing cofactor NADPH. As GSH, Trx is likewise an astroglia-derived neuro-protectant [6,7,8].

The inability of the antioxidant defense system to counteract ROS overproduction in the central nervous system (CNS) is coupled with a lack of energy, seen in all ONS-associated diseases, including MS. Low adenosine three phosphates production (ATP), a vital source of energy, is the consequence of the slaughtered function of mitochondria, as seen in MS [9]. Feedback mechanisms by which glucose-6-phosphate may be utilized in glycolysis to compensate for the lack of energy results in ATP and NADH production used to store energy in the form of glycogen, or used by the pentose phosphate pathway compensates energy loss and keep the viability of cells [10]. Oxidative stress hinders the myelination of axons mediated by oligodendrocytes and oligodendrocyte progenitor cells (OPCs) differentiation into oligodendrocytes [11].

The repetitive transcranial stimulation arouses remyelination and fosters the expression of neurotrophic factors that are of significant importance for neuro-regeneration in EAE animals [12]. In the cerebral cortex, it may promote neuronal activities in remote zones of the spinal cord connected to the primary motor pathway—corticospinal tract. Additionally, neuronal activity is found to be essential for OPCs differentiation into oligodendrocytes in the CNS [11,13].

Structured patterns of repetitive transcranial stimulation, such as theta burst stimulation (TBS), modulate motor cortical excitability. Intermittent TBS (iTBS) provokes an early stage of long-term potentiation, while continuous TBS (cTBS) induces an early stage of long-term depression. Cortical neuronal activities (excitation and inhibition) are associated with neuronal plasticity, support, and remyelination of the affected neural tissue in the lumbar spinal cord [14]. The precise underlying mechanisms of iTBS or cTBS have not revealed yet.

When considering the above data, we aimed to evaluate redox homeostasis in EAE and during iTBS and cTBS in the spinal cord of EAE in rats.

## 2. Materials and Methods

### 2.1. Animals and Experimental Procedure

The experiment was conducted in female Dark Agouti (DA) rats, weighed 150–200 g, and aged 10–14 weeks [15]. The animals (six per cage) were housed in polyethylene cages under standardized housing conditions (ambient temperature of 23 ± 2 °C, the relative humidity of 55 ± 3%, a light/dark cycle of 13/11 h). The access to laboratory food and water was ad libitum, and for welfare assistance to animals with developed EAE, food, and water were bottom-positioned. The rats were habituated to the ambient and laboratory staff five days before the experimental procedure. The experiment lasted 25 days. The TBS started from 14th post-immunization day (dpi) in EAE rats and lasted ten days. Rats were intraperitoneally anesthetized with sodium pentobarbital and decapitated after 24 h of the last treatment [16].

This EAE animal study is a part of the more significant EAE project, approved by the Ethical Community of the Military Medical Academy (Belgrade, Serbia) (license no. 323-07-00622/2017-05), which followed the principles of the governmental policy of Official Gazette Republic of Serbia (No. 14/2009) and Directive 2010/63/EU.

The immunization of rats for EAE was performed by subcutaneous injection of 0.1 mL of rat spinal cord homogenate (50% *w/v* in saline) that was suspended in Complete Freund’s Adjuvant (CFA), containing 1 mg/mL *Mycobacterium tuberculosis* (CFA; Sigma, St. Louis, MO, USA) in the right hind rat footpad. The rats were previously anesthetized intraperitoneally with sodium pentobarbital 45 mg/kg body weight (Trittay, Germany) [16].

Theta burst stimulation by MagStim Rapid2 device with a 25 mm figure-of-eight coil (The MagStim Company, Whitland, Dyfed, UK) was manually performed. The coil was positioned over bregma in direct physical contact with the animal head [14]. The iTBS block consisted of 600 pulses during 192 s. Twenty trains of 10 bursts with three pulses at a frequency of 50 Hz, repeated at 5 Hz, with 10 s pauses between trains were applied [17,18]. The cTBS consisted of 600 pulses set in a single train of bursts during 40 s, repeated at 5 Hz. The applied intensity corresponded to the 30% of maximal strength of TBS, which was merely under a motor threshold of rats, evaluated as a visible contraction of upper limbs without any other apparent distress (Scheme 1). The rats were monitored daily and clinically scored for EAE neurological signs.

### 2.2. Clinical Evaluation

The double-blind approach was applied for the rats’ daily observation and scoring of EAE signs, up to 24 dpi. The clinical symptoms and signs were ranged from 0 to 5: 0—no oddity; 0.5—fairly lose tail tone and inability to rotate the posterior side of the tail; 1—depressed tail tonus; 1.5—moderately unsteady walk and reduced ability to straight-up or their combination; 2—weakness of hind limbs; 2.5—incomplete hind limb paralysis; 3—complete hind limb paralysis; 3.5—complete paralysis of posterior and weakness of the front legs; 4—quadriplegia with breathing effort; and, 5—moribund or death [19].

The following descriptors (presented as Mean ± STDEV) for the severity of EAE clinical symptoms and monitoring of the iTBS or cTBS effects were considered: incidence (proportion of rats with clinical score ≥0.5 per group); daily clinical score (rating of clinical symptoms per rat per day); maximal clinical severity score (duration of the most severe clinical symptoms (days)); EAE onset (the appearance of the early clinical signs (day)); duration of paralysis (period after the EAE onset, when rats had a score ≥2.5 (days)); and, mortality rate.

### 2.3. Tissue Homogenates and Biochemical Analyses

The spinal cords were dissected, and lumbar region slices were transferred separately into saline solution (0.9% *w*/*v*). A crude mitochondrial fraction was prepared, as follows: aliquots (1 mL) were homogenized (homogenizer—Tehtnica Zelezniki Manufacturing, Zalezniki, Slovenia) twice using Teflon pestle at 800 rpm (1000× *g*) for 15 min. at 4 °C, and centrifuged at 2500× *g* for 30 min, at 4 °C. The subcellular membranes pellet was solubilized in 1.5 mL of deionized water, by constant mixing with Pasteur pipette, for 1 h. The mixture was then centrifuged at 2000× *g* for 15 min., at 4 °C, and supernatant (a crude mitochondrial fraction) was used for analysis [20].

Nitrite and nitrate concentration (NO_2_+NO_3_) was spectrophotometrically determined at 492 nm, using the colorimetric Griess method [21].

Lipid peroxidation was spectrophotometrically determined by using the thiobarbituric acid reactive species (TBARS) assay, as described by Girotti et al. The results are expressed as nmol of malondialdehyde (MDA) per milligram of proteins (nmol MDA/mg protein) [22].

Superoxide anion (O_2_^•−^) was spectrophotometrically determined at 550 nm. The principle of the method relates to the reduction of nitroblue-tetrazolium (Merck, Darmstadt, Germany) in the alkaline, nitrogen saturated medium [23].

Superoxide dismutase (EC 1.15.1.1.; SOD) activity was expressed as the degree of inhibition of epinephrine auto-oxidation by SOD, in the presence of O_2_, in alkaline pH. It was spectrophotometrically measured at 480 nm [24].

Total sulfhydryl groups (SH) were spectrophotometrically measured at 412 nm by Elman’s method [25].

Total GSH was spectrophotometrically measured at 412 nm, for 6 min., by 5,5-dithiobis-2-nitrobenzoic acid (DTNB)—oxidized glutathione (GSSG) recyclable method. The level of produced 5-thio-2-nitrobenzoic acid (TNB) was proportional to the total GSH concentration [26]. The content of total GSH was expressed as nmol/mg protein.

Thioredoxin reductase activity was spectrophotometrically measured at 412 nm with a commercially available Assay kit (Abcam; ab83463). The principle of the method refers to the reduction of 5,5′-dithiobis(2-nitrobenzoic) acid (DTNB) into yellow-colored 5-thio-2-nitrobenzoic acid (TNB), catalyzed by thioredoxin reductase that utilizes NADPH. The results are expressed as mU/mg proteins (one unit of thioredoxin reductase activity equals the formation of 1,0 μmol of TNB per minute at 25 °C) [27].

The reduced form of NADPH was spectrophotometrically determined using commercial NADPH Assay Kit at OD450nm (Abcam, Cambridge, UK) [28].

Total proteins were determined by the Lowry method [29].

### 2.4. Statistical Analysis

One-way ANOVA and Bonferroni’s post hoc multiple tests were used (software GraphPad Prism, version 5.03) for statistical data analysis. The values are presented as means ± SD. The level of confidence referred to *p* < 0.05.

## 3. Results

### 3.1. Clinical Observation

The onset of EAE symptoms was established at 10.5 ± 0,23 dpi in all immunized rats (100% incidence of the EAE onset). No lethality occurs in any of the experimental groups. The same pattern, addressing the onset and the duration of EAE clinical manifestations, was recognized within the non- and TBS treated EAE groups. Non-TBS treated groups refer to EAE and sham i/cTBS treated EAE groups, and TBS treated EAE groups refer to i/cTBS treated EAE groups. The duration of limb paralysis (*p* < 0.05) and severity of EAE symptoms were lower in TBS treated EAE rats than in EAE rats (Table 1).

The progress of EAE (development and withdrawal of severe clinical signs) overlapped with body mass loss and its gaining back (Figure 1).

The drop in the body mass and gaining back was more intense in non-TBS treated than in TBS treated EAE rats (reduction for 30% and 19%; and, achieved 96% vs. 90% of the initial weight, respectively) as compared to 10th dpi, when EAE onset happened. The body mass loss coincided with paralysis (Table 1, Figure 1).

Clinical symptoms, including motoric/walking possibilities and developed inabilities, were observed daily and scored in EAE, TBS-, and sham-TBS- treated EAE rats. The totals of daily clinical scores were summarized for days before (0–13th dpi) and during (14–24th dpi) the applied TBS or sham TBS to EAE rats (Figure 2). The summarized clinical scores for 14–24th dpi were significantly lower in TBS treated EAE rats when compared to EAE rats (*p* < 0.05), whereas cTBS achieved better health effects.

### 3.2. Oxidative Stress Biochemical Analyses

The concentrations of NO_2_+NO_3_ increased in the TBS-untreated groups (EAE and EAE + iTBSsh and EAE + cTBSsh) (*p* < 0.05) compared to both iTBS and cTBS-treated EAE groups, which remained equal to the control values (Figure 3a).

The concentrations of TBARS increased with progress and peak of the disease (EAE14; *p* < 0.01), but lately it decreased (EAE24; *p* < 0.01). Decreased TBARS values were documented in both TBS-treated EAE groups (EAE + iTBS and EAE + cTBS) and TBS-untreated EAE groups (EAE + iTBSsh and EAE + cTBSsh), when compared to EAE14 (*p* < 0.01) (Figure 3b).

In both of TBS-treated (EAE + iTBS, EAE + cTBS) and TBS-untreated EAE groups (EAE + iTBSsh, EAE + cTBSsh), the production of O_2_^−^ decreased, as compared to the peak of the disease (EAE14; (*p* < 0.01). In this term, O_2_^−^ elevated, when compared to controls (*p* < 0.001). Remarkable data were depleted production of O_2_^−^ in healthy animals treated with iTBS (*p* < 0.05) and cTBS (*p* < 0.01), compared to controls (Figure 3c).

A significant drop of SOD activity was measured on the 10th day in EAE rats (EAE10; *p* < 0.001) as compared to the controls, while afterward, it started to rise (EAE14/24, *p* < 0.01; compared to EAE10). The activity of SOD also decreased in CFA treated animals (*p* < 0.001), cTBS-treated EAE group (EAE + cTBS; *p* < 0.001) and in TBS-nontreated rats (EAE + iTBSsh and EAE + cTBSsh; *p* < 0.001). Increased SOD activity occurred in EAE + iTBS rats, as compared to all EAE groups (EAE10, *p* < 0.001; EAE14, *p* < 0.01; EAE24, *p* < 0.05) (Figure 3d).

The concentration of thiols (SH-containing compounds) declined with the disease progress (EAE14, *p* < 0.05: compared to the controls). Elevated values of thiols were measured in both iTBS and cTBS treated EAE compared to all EAE groups (EAE10/14/24 groups, *p* < 0.05) (Figure 4a).

The GSH decreased in EAE10 and EAE14 groups compared to the controls (*p* < 0.05). The renewal of GSH content happened on the 24th day (EAE24). The cTBS increased GSH in EAE rats (EAE + cTBS), when compared to EAE24 (*p* < 0.05). Interestingly, iTBS increased GSH in healthy animals (*p* < 0.001) (Figure 4b).

The activity of TrxR increased in all experimental groups [CFA (*p* < 0.05); EAE10,14,24 (*p* < 0.01); EAE + i/cTBS (*p* < 0.01); EAE + i/cTBSsh (*p* < 0.01); i/cTBS (*p* < 0.01)] compared to controls. The activity of TrxR decreased in TBS-untreated groups [(EAE + iTBSsh (*p* < 0.05) and EAE + cTBSsh (*p* < 0.05)] as well as in EAE + cTBS rats (*p* < 0.05), as compared to EAE24 group (Figure 4c).

The depletion of NADPH occurred in the spinal cord of EAE 10/14/24, CFA group, and both sham TBS-treated EAE rats, compared to the controls (all *p* < 0.05). The iTBS and cTBS increased NADPH concentration back to control levels in treated EAE rats. Additionally, NADPH in EAE + iTBS rats was higher from EAE10/14/24 groups (*p* < 0.05) (Figure 4d).

## 4. Discussion

Herein, we provided inclusive results on the status of ROS/RNS, reducing equivalents (NADPH and thiol compounds, including GSH) and relevant antioxidant enzymes, such as SOD and TrxR in rats immunized for EAE and during iTBS and cTBS. The peak of ONS coincided with the climax of EAE clinical manifestation. Besides, we will discuss the experimental and clinical achievements of the therapeutic protection in the pathogenesis of EAE, i.e., MS [30].

The close association of EAE pathology and ONS was confirmed by the overlapping of the peak of EAE clinical score, achieved at 14 dpi and highpoint of ONS measured in the spinal cord of treated rats. Among the measured ONS parameters, the focus fell on TrxR, which appears to be the most sensitive parameter of ONS [31]. Additionally, we documented that TBS of EAE rats mitigated the ONS in the spinal cord and improved clinical symptoms of EAE. The effect was realized, both at the site of stimulation and in the zone of the spinal cord [32,33]. Contrary to the report on opposite modulation directions of cortical excitability by cTBS and iTBS in EAE, we obtained almost similar outcomes by both iTBS and cTBS, with somewhat more beneficial effects being accomplished by cTBS (Figure 2).

Underlying mechanisms of TBS to raise neuronal activity, adult neurogenesis, and remyelination mediated by astrocytes and microglia/macrophages have not been fully elucidated. The lack of pharmacological treatment effectiveness fortifies the exploitation of nonpharmacological and noninvasive repetitive transcranial magnetic stimulation for cognitive rehabilitation in MS patients. This neurostimulatory and neuromodulatory technique realize electromagnetic induction of an electric field in the brain, with behavioral consequences and therapeutic potentials. Locally and remotely, it affects neural functions, excitatory, or inhibitory [34].

In terms of nitrosative stress, our results showed that NOx levels were the highest in EAE rats, while lately, applied TBS reduced it slightly (Figure 3a). Nitrogen monoxide regulates cerebral blood flow and neuronal cell viability. As a weak oxidant, NO is incapable of inducing ONS, unlike its radical form (NO·), which causes only modest BBB disruption, while in combination with other RNS elicits severe BBB damage. Additionally, high levels of RNS in the brain participate in neurotoxicity [35]. Increased metabolism of NOx is the hallmark of glutamate turnover, i.e., excitotoxicity in EAE. Glutamate-mediated excitotoxicity occurs via ionotropic glutamate receptors activation and increased calcium (Ca^2+^) influx. Cells Ca^2+^ overload affects mitochondrial respiratory chain and ATP production, which is an early occurrence in EAE lesions.

Additionally, macrophages and microglia influence metabolic and regenerative processes, striping myelin and inducing matrix metalloproteinases that are involved in the vascular response to oxidative insult [35]. The localization of mitochondria within axons is essential for maintaining axonal integrity, demyelination, and remyelination [36]. The release of ROS from macrophages occurs during phagocytosis (“oxidative burst”) of engulfed debris and toxic products from the surrounding environment [37]. Oxidatively damaged neighbor health tissue, although not affected by inflammation, can facilitate the infiltration of immune cells, which is the scenario recognized in an early stage of EAE [38]. Assumingly, that could be the explanation of the simultaneous occurring of EAE symptoms severity and climax of ONS.

Antioxidant enzyme SOD converts O_2_^•−^ into molecular oxygen and hydrogen peroxide, by utilizing NAD(P)H. Reduced SOD activity, documented on 10 dpi, was followed with the reduced elimination of O_2_^•−^, which resulted in its peak on 14 dpi, which coincided with the severity of lipid peroxidation (expressed as TBARS) and the most severe clinical signs in EAE rats. However, the return of SOD to normal on 14th dpi resulted in the consequent decrease of O_2_^•−^ and TBARS to control values (Figure 3b,c), according to the study of Thimm et al. (Figure 3c,d) [39]. Proapoptotic O_2_^•−^ emerges tumor necrosis factor-α (TNF-α) mediation in ONS in the cerebral vascular endothelium. It is known that TNF-α has a pivotal role in various types of inflammatory brain diseases. Consequently, the removal of O_2_^•−^ reduces inflammation via the TNF-α pathway [35].

Reduced SOD activity along with NADPH in CFA, EAE, and sham-TBS treated EAE rats (Figure 3d) indicate that both nonspecific immuno-stimulation (CFA-mediated) and specific (immunization for EAE) deteriorated redox homeostasis in the direction of ONS [40]. The distinctive impact of iTBS and cTBS on SOD activity was documented in EAE treated rats. Its activity increased under the iTBS, while cTBS did not impose any effect. The nuclear factor (erythroid-derived-2)-like 2 factor (Nrf2) upregulates antioxidative enzymes in nervous tissue. The Nrf2 path quenched in EAE; thus, we speculated that iTBS supports the Nrf2 pathway that supports SOD activity [41]. Mitochondrial function is particularly compromised by the excess of glutamate and pro-inflammatory mediators (proteases, inflammatory cytokines, etc.). The oxidative damage of neuronal and glial cells, including OPCs, as well as recruited T cells at the site of inflammation, could be a reason for matrix metalloproteinases production and the blood-brain barrier disrupter. The dysregulation of the BBB is among the earliest cerebrovascular abnormalities seen in MS brains. Deteriorated BBB integrity increases the permeability of cerebral micro-vessels seen in many neurological inflammatory diseases [42].

The occurrences mentioned above coincided with a decrease of GSH, total sulfhydryl (-SH) containing compunds (thiols), and NAD(P)H (Figure 4a,b,d). The changes in total SH-containing compounds and GSH were similar (Figure 4a). It is well-known that -SH groups of cysteine in GSH is responsible for its antioxidant and other physiological roles. As a strong nucleophile, SH-groups are prone to oxidation, which was probably the reason for the reduced content of thiols and GSH documented in EAE and EAE + iTBSsh or EAE + cTBSsh groups, as compared to controls [43]. Thiols reduce an oxidative attack and preserves surrounding biomolecules from oxidative damage. The functional impairment of SH-groups containing molecules could arise from metabolic disorders but also long-lasting overstimulation of neurons in EAE. In excitotoxicity, an overload of the glutamate-cysteine transporters blocks cysteine transport, affecting cysteine, which is the source of SH-group in GSH, and other thiol-proteins [44]. The GSH cycle encompasses several antioxidant enzymes that utilize GSH or NADPH as donors for reducing equivalents.

Nevertheless, if applied to EAE or health rats, both iTBS and cTBS return the thiols and GSH to the level controls. Mitochondrial GSH is recognized to be a critical factor in antioxidant protection. Thus, impairment of mitochondrial physiology may considerably ruin the GSH antioxidant capacity [45]. The antioxidant effect of thiols may be the primary underlying mechanism of ROS sequestration during reactive astrogliosis treated with TBS, particularly by iTBS, which arrives more effective than cTBS in preserving/restoring of the GSH reserves (documented in health rats) (Figure 4b). The use of transcranial magnetic stimulation in human medicine has been increasing lately, although limited evidence of its effect on glia in vivo is available nowadays [46].

Unlikely to the GSH antioxidant role, little is known regarding the physiological significance of the Trx system in this capacity [5,47,48,49]. To our knowledge, the present study is the first in the literature to address the role of the Trx/TrxR system relative to the GSH system in the response of the brain to induce EAE in rats.

Down-regulated antioxidant enzymes, next to impaired mitochondrial GSH pool in EAE rats (Figure 4b), imply that GSH may not be effective enough in coping with the ONS evoked by EAE [45]. Contrary to the reduced ROS sequestrating system (herein represented by SOD) and GSH system, we obtained a remarkable increase of TrxR in rats that were immunized for EAE (specific immunization), CFS (nonspecific vaccination), and TBS-treated health rats (Figure 4c). The decreased level of total thiols (Figure 4a) correlates with increased TrxR activity. By that, we confirmed the exceptional protection role of TrxR against brain oxidant injury in rats and its outstanding sensitivity to the applied CNS stressors, according to the Ruszkiewicz, J. and Jan Albrecht, J. study [45]. It appears that the increase of TrxR activity accomplishes a neuroprotective role. That is the positive feedback mechanism against ONS. We showed that upregulated TrxR compensated reduced GSH and SOD antioxidant role in EAE rats, underlying its antioxidant role (Figure 4c). No differences in TrxR activity were documented between EAE and healthy rats that were subjected to TBS. However, the capacity of the Trx/TrxR system in different neurological diseases, such as EAE or during TBS, is not clarified yet. Thioredoxin reductase imposes additional activities, besides the reduction of Trxs. It recycles ascorbate and imposes another biological role in cell growth and transformation. However, increased brain Trx/TrxR role was seen in hyperoxia in newborn rats and ischemia induced by transient middle cerebral artery occlusion in adult mice, and in Alzheimer’s disease-affected brain [50,51,52]. The Trx/TrxR system accomplishes its neuroprotective antioxidant role in the brain. It appears that the feedback mechanism of the Trx/TrxR system supports the overcoming of the ONS-associated astrogliosis in EAE [12,53,54].

The decrease of NADPH in EAE rats was expected, due to its cofactor role in numerous enzymatic oxidoreductive reactions, including GSH turnover by GR, reduction of Trx; and, other SH-compounds, prone to oxidation (Figure 4d) [55]. The obtained result confirmed the increased utilization of NADPH during OS and NS in EAE rats. As a cofactor in lipid and cholesterol synthesis and fatty acid chain elongation, NAD(P)H is vital for neuronal tissue maintenance [56]. The consumption of GSH and NAD(P)H in enzymatic reactions of FRs’ sequestrations entails the involvement of the pentose phosphate pathway [57]. The recovery of NADPH and total thiols and glutathione under the iTBS or cTBS promotes better metabolic achievement in EAE. No statistical significance of TrxR activity between EAE or healthy rats subjected to TBS was obtained.

The maintenance of redox homeostasis and energy is a predominant target for EAE treatment. The peak of EAE clinical symptoms positively correlated with the climax of ONS, including the depletion of donors of reducing equivalents (NADPH and total thiols, including GSH), fall of SOD activity, but the rise of TrxR activity. We recognized the upregulation of TrxR as a feedback mechanism against red-ox balance perturbation in the brain and spinal cord of rats induced by stressors, such as specific immunization for EAE, nonspecific immunization with CFS, and TBS as well.

A similar effect of TBS on TrxR activity was achieved in EAE and healthy rats. That is difficult to explain because of the lack of knowledge addressing the Trx/TrxR system role in neurological inflammatory diseases. iTBS and cTBS both modulate the biochemical environment against spinal cord ONS and alleviate symptoms of EAE. Both TBSs shifted neuronal antioxidative defense towards a more reductive state to improve physiological resilience to ONS by modulating the biochemical environment in EAE at a distance from the area of stimulation. iTBS and cTBS both modulate the biochemical environment against ONS at a distance from the area of stimulation, alleviating symptoms of EAE.

This animal study significantly increases our understanding of FRs’ interplay and the role of Trx/TrxR in ONS-associated neuroinflammatory diseases, such as EAE. For now, the antioxidant-based therapies and management of neurodegenerative diseases in clinical and experimental settings have identified antioxidant based therapies to be largely ineffective [58].

The results of our study might help the development of new ideas for designing more effective medical treatment, combining neuropsychological with noninvasive neurostimulation—neuromodulation techniques for patients living with MS [34].

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
