# Peer review of "Compensatory Neuroprotective Response of Thioredoxin Reductase against Oxidative-Nitrosative Stress Induced by Experimental Autoimmune Encephalomyelitis in Rats: Modulation by Theta Burst Stimulation"

_molecules, 2020, doi:10.3390/molecules25173922_

Round 1

Reviewer 1 Report

The authors presented data about how iTBS or cTBS affect the symptoms of EAE model animals. Their overall research is adequately conducted. The manuscript could be better if the following issues are solved.

L20-23 Describe what is known and what you are going to show in this study more specifically.
L23-27 This part looks like a list of abbreviations.
L27-32 Describe what is the most critical result you get from this study, instead of a list of analysis you performed.
L32-34 Describe the results that support this conclusion and the logic behind it.

L172 Describe what is used as criteria of EAE onset, that scored 100%.
L175-176, Table 1, The definition of each group is hard to understand. What is the difference between “refer to EAE and sham i/cTBS treated EAE groups” and “refer to i/cTBS treated EAE groups”? It may help to provide a figure panel to explain the difference.

I cannot see which part is Figure legends and which part is the main body. Some section in the main body looks like Figure legends, which is confusing to the readers.

Figure 2 Error bars should be provided. Why the scores of two different time periods are added in this bar graph?

Figure 3,4 When the samples except EAE series are processed? Why the time point is chosen?

Author Response

 (Reviewer 1)

The authors are grateful for the criticism, which helped us to make discussion substantially better. Please find bellow the made changes:

L20-23 Describe what is known and what you are going to show in this study more specifically.

We changed into:

The interplay of brain antioxidant defense systems against free radicals (FRs) overproduction induced by EAE, as well as during iTBS or cTBS have not been entirely investigated. This study aimed to investigate if oxidative-nitrogen stress (ONS) is one of the underlying pathophysiological mechanisms of EAE which may be changed in terms of health improvement by iTBS or cTBS.

L23-27 This part looks like a list of abbreviations.

We changed into:

The rats were randomly divided into the control group, rats specifically immunized for EAE and nonspecifically immuno-stimulated with Complete Freund's adjuvant. TBS or sham TBS were applied to EAE rats from 14th-24th post immunization day.

L27-32 Describe what is the most critical result you get from this study, instead of a list of analysis you performed

We changed into:

The severity of EAE clinical coincided with the climax of ONS. The most critical result refers to TrxR, which immensely responded against the applied stressors of central nervous system (CNS), including immunization and TBS.

L32-34 Describe the results that support this conclusion and the logic behind it.

We changed into:

Compensatory neuroprotective accomplishment of the TrxR upregulation may be interpreted as a positive feedback mechanism against brain ONS induced by stressors, such as immunization for EAE (specific), with CFS (unspecific), and TBS. Both, iTBS and cTBS modulate the biochemical environment against ONS at a distance from the area of stimulation, alleviating symptoms of EAE. Similar effect on TBS to TrxR of EAE and healthy rats is difficult to explain, due to unreliable data on Trx/TrxR role in neurological inflammatory diseases.

L172 Describe what is used as criteria of EAE onset, that scored 100%.

The explanation:

Sorry, as it was written, obviously was confusing for the reader. The criteria of EAE symptoms which were scored are explained in the section Material and Methods [Lines: 121-126]. The point of the sentence [Line 173] was that EAE onset (no matter of the intensity - the score of the clinical symptoms) occurred in all animals [100% incidence of the EAE onset]. 

So, the change what was made relates to:

[100% incidence of the EAE onset]. 

L175-176, Table 1, The definition of each group is hard to understand. What is the difference between “refer to EAE and sham i/cTBS treated EAE groups” and “refer to i/cTBS treated EAE groups”? It may help to provide a figure panel to explain the difference.

The changes are made in the Table 1 and the text of the legend is inserted:

Legend: EAE - rats with experimental autoimmune encephalomyelitis; EAE+i/cTBS [EAE+iTBS, and EAE+cTBS groups – EAE rats subjected to iTBS and cTBS]; EAE+i/cTBSsh [EAE+iTBSsh and EAE+cTBSsh – EAE rats subjected to iTBS's and cTBS's noise artifact (sham-treated EAE rats)].

I cannot see which part is Figure legends and which part is the main body. Some section in the main body looks like Figure legends, which is confusing to the readers.

Sorry, it was technical failure during full text formatting. It was corrected.

Figure 2 Error bars should be provided. Why the scores of two different time periods are added in this bar graph?

The error bars are incorporated. The chart type is changed. Legend is added. The idea of the summarizing clinical scores for two different periods 0-13 dpi and 14-24 dpi rose from the idea to emphasize the health beneficial effect of TBS compare to EAE rats which was not subjected to TBS (TBS started from 14th post-immunization day (dpi) in EAE rats and lasted ten days).

The following text Lines 201-203: clarified the purpose of this graphical presentation:

Lines 201-203: The summarized clinical scores for 14-24th dpi were significantly lower in TBS treated EAE rats compared to EAE rats (p < 0.05), whereas cTBS achieved more intense beneficial health effect.

Figure 3,4 When the samples except EAE series are processed? Why the time point is chosen?

We chose that time point according to the literature: reference 16 [Lavrnja, I. et al], and added the refeerence.

Lines 89-91: Rats were intraperitoneally anesthetized with sodium pentobarbital and decapitated after 24 hours of the last treatment [16].

The other changes that we made in the MS are highlighted in yellow in the attached revised MS.

Reviewer 2 Report

The authors of this study are trying to show the positive effects of theta burst stimulation in EAE. This is a relatively well presented study, although it can be improved. 

  1. Please discuss current advances in MS using TBS. I felt that a significant part of the discussion focused on issues that the authors do not even examine, but lacked a connection to human therapeutics, which would explain why this study is significant. What are the advantages in clinical research? Please discuss
  2. The authors do not discuss at all the effects of TBS on blood brain barrier integrity and how BBB leakage observed in EAE could affect their results.
  3. This study would be far more interesting if the authors performed immunohistochemistry to look at the effects of TBS (both iTBS and cTBS) on astrocytes, microglia and/or myelin and axonal intensity. 

Author Response

 (Reviewer 2)

The authors are grateful for the criticism, which helped us to make discussion substantially better. Please find below the made changes:

  1. Please discuss current advances in MS using TBS. I felt that a significant part of the discussion focused on issues that the authors do not even examine, but lacked a connection to human therapeutics, which would explain why this study is significant. What are the advantages in clinical research? Please discuss

Since the presented results are part of a much wider investigation, we made the omission, writing about the things that the presented results did not refer to.

We have made the following changes:

  1. removed part of the text from the beginning of the Discussion

Subtle pathological changes occur in the brain or spinal cord of rats immunized for EAE before the onset of EAE signs. Astrogliosis has been reported as an underlying mechanism of EAE [12]. Gliosis is usually established after two to three weeks of a CNS injury in rats, which

  1. The new text was incorporated into the MS (along with new references) to follow the Reviewer 2 requirements:

Lines: 36-40 The result of this study increases our understanding of FRs´ interplay and the role of Trx/TrxR in ONS-associated neuroinflammatory diseases, such as EAE and might be helpful to develop new ideas for designing more effective medical treatment, combining neuropsychological with noninvasive neurostimulation–neuromodulation techniques to patients living with MS.

Lines: 288-297 Herein, provided inclusive results on the status of ROS/RNS, reducing equivalents (NADPH and thiol compounds including GSH) and relevant antioxidant enzymes such as SOD and TrxR, during neuroinflammation induced by the immunization of rats for EAE and the applied iTBS and cTBS suggested that harmful effects of ONS coincided with the clinical manifestation of EAE in rats. In addition, we will discuss experimental and clinical achievements of the therapeutic protection in the pathogenesis of EAE, i.e. MS [30] . Close association of EAE pathology and ONS is confirmed by the overlapping of the peak of EAE clinical score, achieved at 14dpi and highpoint of ONS measured in the spinal cord of treated rats. Among the measured ONS parameters, the focus fell on TrxR, which appears to be the most sensitive parameter of ONS [31].

Lines: 305-310 However, the utilization of nonpharmacological and noninvasive interventions such as repetitive transcranial magnetic stimulation for cognitive rehabilitation in MS patients has been fortified by the lack of pharmacological treatment effectiveness. This neurostimulatory and neuromodulatory technique realize electromagnetic induction of an electric field in the brain, with behavioral consequences and therapeutic potentials. Locally and remotely it affects neural functions, excitatory, or inhibitory [34].

  1. The authors do not discuss at all the effects of TBS on blood-brain barrier integrity and how BBB leakage observed in EAE could affect their results.

The texts below were incorporated into the MS (along with new references) to follow the Reviewer 2 requirements:

Lines: 312-316 Nitrogen monoxide regulates cerebral blood flow and neuronal cell viability. As a weak oxidant, NO is incapable to induce ONS, unlike its radical form (NO·) that causes only modest BBB disruption, while in combination with other RNS elicits severe BBB damage. Also, high levels of RNS in the brain participate in neurotoxicity [35].

Lines: 336-339 Proapoptotic O2•- emerges tumor necrosis factor-α (TNF-α) mediation in ONS in the cerebral vascular endothelium. It is known that TNF-α has a pivotal role in various types of inflammatory brain diseases. Consequently, the removal of O2•- reduces inflammation via the TNF-α pathway [35].

Lines: 352-355 Dysregulation of the BBB is among the earliest cerebrovascular abnormalities seen in MS brains, which. Deteriorated BBB integrity increases permeability of cerebral micro-vessels seen in many neurological inflammatory diseases [42].

Lines: 379-381 The use of transcranial magnetic stimulation in human medicine has been increasing lately, although limited evidence of its effect on glia in vivo is available nowadays [46]

  1. This study would be far more interesting if the authors performed immunohistochemistry to look at the effects of TBS (both iTBS and cTBS) on astrocytes, microglia, and/or myelin and axonal intensity.

We agree with you. We did it a wider range of the investigation on EAE and applied TBS in rats, but herein we presented and discussed the results that addressing only oxidative-nitrosative stress. The results addressing immunohistochemistry of the glial glutamate transporters, that we also investigated are presented in another MS, which is currently under the review in the journal Scientific Reports.

The other changes that we made in the MS are highlighted in yellow in the attached revised MS.

Round 2

Reviewer 2 Report

No additional comments